# Local stimulation of osteocytes using a magnetically actuated oscillating beam

Onaizah Onaizah[1☉], Liangcheng Xu[2☉], Kevin Middleton[2], Lidan You[1,2]*, Eric Diller[1]

1 Department of Mechanical and Industrial Engineering, University of Toronto, Ontario, Toronto, Canada,
2 Institute of Biomaterials and Biomedical Engineering, University of Toronto, Ontario, Toronto, Canada

☉ These authors contributed equally to this work.
* youlidan@mie.utoronto.ca

**Data Availability Statement:** All relevant data are within the paper and its Supporting Information files

**Funding:** The author(s) received no specific funding for this work.

## Abstract

Mechanical loading on bone tissue is an important physiological stimulus that plays a key role in bone growth, fracture repair, and treatment of bone diseases. Osteocytes (bone cells embedded in bone matrix) are well accepted as the sensor cells to mechanical loading and play a critical role in regulating the bone structure in response to mechanical loading. To understand the response of osteocytes to differential mechanical stimulation in physiologically relevant arrangements, there is a need for a platform which can locally stimulate bone cells with different levels of fluid shear stress. In this study, we developed a device aiming to achieve non-contact local mechanical stimulation of osteocytes with a magnetically actuated beam that generates the fluid shear stresses encountered *in vivo*. The stimulating beam was made from a composite of magnetic powder and polymer, where a magnetic field was used to precisely oscillate the beam in the horizontal plane. The beam is placed above a cell-seeded surface with an estimated gap height of 5 μm. Finite element simulations were performed to quantify the shear stress values and to generate a shear stress map in the region of interest. Osteocytes were seeded on the device and were stimulated while their intracellular calcium responses were quantified and correlated with their position and local shear stress value. We observed that cells closer to the oscillating beam respond earlier compared to cells further away from the local shear stress gradient generated by the oscillating beam. We have demonstrated the capability of our device to mimic the propagation of calcium signalling to osteocytes outside of the stimulatory region. This device will allow for future studies of osteocyte network signalling with a physiologically accurate localized shear stress gradient.

## Introduction

Mechanotransduction is an important process for basic cell functions, affecting key cellular mechanisms such as protein signalling and DNA transcription. Physical cues act as fundamental inputs to mechanotransduction, ranging from mechanical stimulation of the cell surface to unique physical properties embedded in the surrounding extracellular matrix [1,2]. Although observed in a variety of organ systems, these physical cues are most prominent in load-bearing tissues such as the bone.

**Competing interests:** The authors have declared that no competing interests exist.

In bone tissue, osteocytes, the major mechano-sensory cells, are embedded within the lacunar-canalicular network exposing them to high levels of fluid shear stress upon bone tissue compression [3]. This mechanical stimulus is important for bone tissue function, as it activates key signalling pathways that regulate the bone remodelling process [4,5]. Osteocytes seeded within *in vitro* fluid flow systems have demonstrated their sensitivity to different levels of fluid shear stress [6–8]. However, current typical *in vitro* systems rely on macro-scale devices that stimulate a monolayer cell culture with uniform shear stress, in contrast with the pockets of shear stress gradients experienced by osteocytes in the lacuna-canaliculi network [9,10]. The rise of microfluidic systems has filled this gap by introducing cell culturing platforms with dimensions closer to that of the lacunar-canalicular network, therefore demanding development of newer fluid stimulation mechanisms beyond traditional parallel flow chambers to mimic more physiologically accurate mechanical stimulation of osteocytes.

One of the early stage cell response to flow in the form of intracellular calcium fluctuations have been successfully detected from both osteocytes cultured using *in vitro* fluid flow systems [11–13], as well as *in vivo* models [14–16]. These calcium flux were measured with either the average response from a population of osteocytes or the single-cell calcium fluctuation pattern. However, it is still very difficult to differentiate the cellular responses that result from intercellular signalling transport from mechanically stimulated cells. Both *in vitro* experiments using patterned cell networks [17] and *ex vivo* studies using bone tissue [18] have demonstrated the key role calcium fluctuations play in propagation of signals between mechanically stimulated and non-stimulated osteocytes; however these studies rely on membrane disturbance and tissue strain as the mechanical stimulus, lacking the capability to study how fluid shear stress influences this type of signal propagation. Existing tools such as atomic force microscopy (AFM) can only provide point-force membrane disturbances to the cell and lack the capability to generate localized fluid shear stress representative of the different levels of shear stress experienced by osteocytes within the lacuna-canaliculi network. Hence there is a need for the development of a platform to locally stimulate a selected region of osteocyte culture with a shear stress gradient to measure the varying response of osteocytes to mechanical stimulation, as well as response from intracellular signalling to non-stimulated cells.

While local stimulation of cells has been attempted in the past [19], no study has attempted to quantify the shear stress gradient that can be generated through local non-contact cell manipulation. Contact cell manipulation involves direct physical contact between the cell wall and external tools or forces whereas non-contact cell manipulation is the indirect manipulation of cells through remote actuation methods such as magnetic fields [20], acoustics [21–23], fluid forces [24,25], and dielectrophoresis [26–28]. Contact manipulation allows for precise control of single cells using techniques such as optics or optical tweezers [29–31], magnetic fields [32–34] or bio-actuation [35,36]. These techniques can be embedded into a microfluidic chip and allow for cutting, injecting and stimulation of individual cells with a high degree of selectivity and accuracy [37–39]. Non-contact cell manipulation has the capability to manipulate a larger number of cells in a multitude of environments but loses some of the accuracy and specificity of contact manipulation. They are, however, less likely to damage cells as a result. Magnetically actuated microrobotic tools have been used in the past for cell manipulation tasks such as transportation [20,40] or proposed for other biomedical applications [41].

This study aims to design a platform which can enable local cell mechanical stimulation by fluid shear stress in a targeted region. A magnetically-actuated beam is placed above adherent MLO-Y4 osteocyte-like cells and oscillated at a frequency of 1 Hz in order to apply fluid shear stresses to the cells. The shear stress is localized to the region surrounding the beam, while cells further away from the stimulated region experience minimal shear stress. Finite element simulations are performed in order to quantify the shear stress values that can be generated by

the oscillating magnetic beam. An experimental protocol is established with a specifically designed coil system and driving electronics integrated into an optical inverted microscope. Live imaging of intra-cellular calcium fluctuations is used to quantify cell response during magnetic actuation. A shear stress map is plotted along with the locations of all stimulated cells in order to illustrate the working principles of the device and to understand how future studies with local cell stimulation can be performed more reliably.

## Methods

### a) Design and fabrication

Local stimulation of cells is achieved through the placement of a magnetically-actuated flexible beam above the adherent cell surface, which can generate localized shear stress regions [42]. The beam is manufactured as a flexible polymer with magnetic material embedded inside. It consists of a mixture of polydimethylsiloxane (PDMS, Sylgard 184, Dow Corning) which comes in two parts with a polymer base and curing agent that are combined in a 10:1 ratio by mass. This mixture is then combined with permanent magnetic particles (MQFP-15-7, NdPrFeB, Magnequench) in a 1:1 mass ratio. This mixture is poured into a negative mold of the beam that was created using photolithography. The excess is scraped off with a razor blade. This mixture is fully cured on an 85˚C hot plate for 4 hours and the beam is subsequently removed from the mold using a needle. After the beam has been removed, the magnetic particles inside the beam are magnetized before assembly by placing the beam in a uniform magnetic field of 1.1 T created by two permanent magnets (1-inch cube, NdFeB, N40, Magnet4US) placed 3 mm apart (Fig 1).

The beam is then glued using liquid PDMS to a glass slide with a spacer added to prevent the sinking of the beam. This is again cured on an 85˚C hot plate for 4 hours. After curing, the spacer is removed. Separately, an adhesive film is added as a border to another glass slide on which cells are cultured. The slide with the beam is then flipped and placed on top of the cell surface. A small gap between the beam and the cell surface is necessary to ensure non-contact manipulation. Two different types of adhesives films are used to leave a gap height of 10 and 25 μm respectively. However, we will show in our results that cell stimulation was likely not achieved until the gap height was 5 μm as shear stress levels at higher gap heights are not sufficient. The small gap heights are achieved coincidentally in these set of experiments, but future experiments can be designed to repeatedly achieve this gap height. The most common reasons for the experimental gap height to be smaller than the theoretical gap height is because of gravity, which pulls the tip of the beam down when the device is assembled or a thick layer of glue that pushes the whole structure downward (see Fig 2). The former would also result in an uneven shear stress gradient resulting in cells closer to the tip of the beam to be more easily stimulated due to higher shear stress levels while the latter would increase the shear stress uniformly across the localized region.

### b) Experimental design

A pair of electromagnetic coils (Fig 3) were designed to fit around a fluorescence microscope. A large set of coils were designed with a 17 cm radius, 300 turns of a 7 AWG copper wire mounted on a wooden structure that generates 10 mT in the center of the workspace. The device is placed in the center of the workspace such that the magnetization direction of the beam is perpendicular to the external magnetic flux density. An oscillating magnetic flux density of 10 mT at 1 Hz (Fig 4B) causes the beam to oscillate in the x-y plane [42] that results in fluid shear stress on the cell surface. From previous literature, commonly accepted range of fluid shear stress sensed by osteocytes in the lacuna-canaliculi network is estimated to be around 0.8–3 Pa [9], with our previous work showing a response from MLO-Y4 osteocytes at a minimum of 0.5 Pa [6]. Hence, the magnitude of the shear stress must be greater than 0.5 Pa to

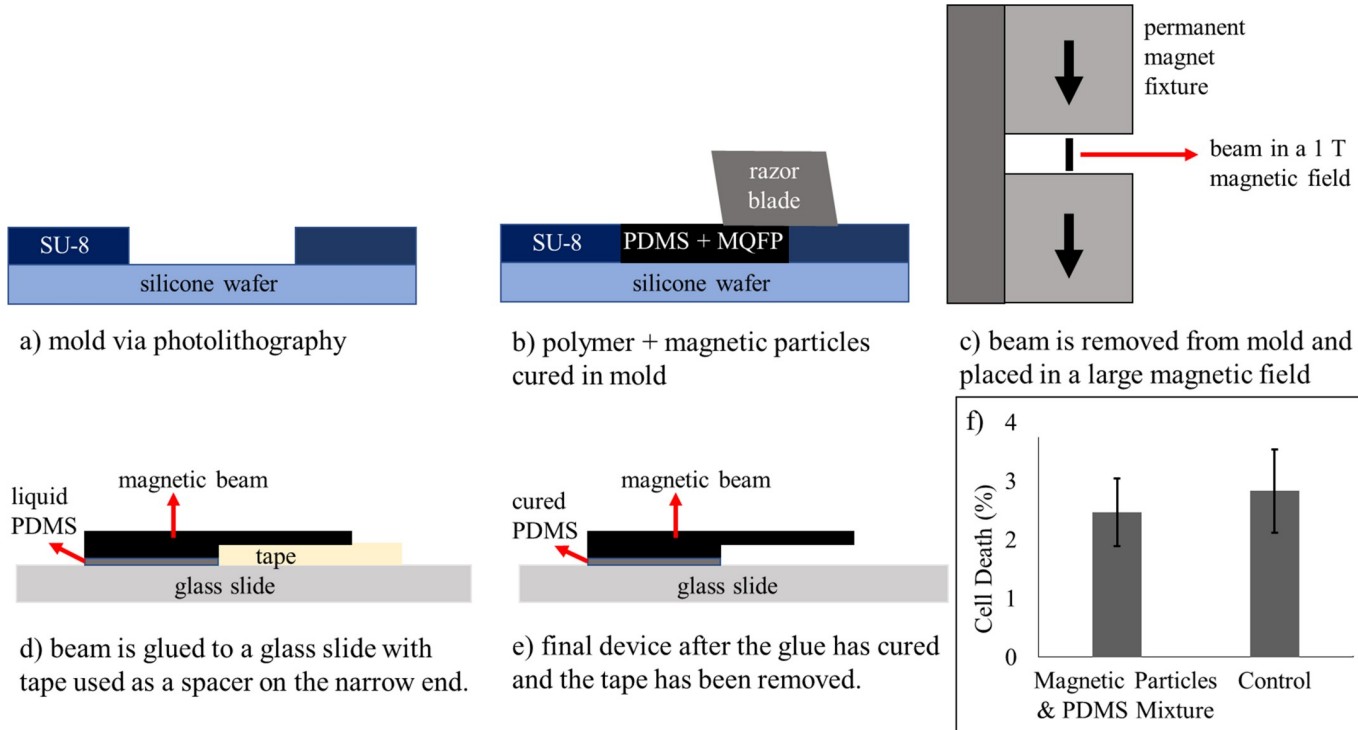

**Fig 1. The fabrication process for the magnetically-actuated beam.** a) a negative mold for the beam is created via photolithography, b) A mixture of PDMS with magnetic particles is cured in the mold with any excess removed via a razor blade, c) the beam is removed after curing and magnetized in a large magnetic field generated by two permanent magnets, d) the magnetic beam is then glued to a glass slide using liquid PDMS and tape as a spacer and again cured, e) the final device when the spacer is removed after the device has been fully cured. f) Experiment demonstrating that the magnetic mixture has no effect on cell death compared to conventional plastic cultureware. N = 3.

result in cell stimulation, which occurs for very small beam-surface gap heights. The coil system is connected to an analog servo driver (30A8, Advanced Motion Controls) and power supply for tunable field generation. A signal generator is used to generate a 1 Hz sinusoidal waveform. The sinusoidal waveform was chosen because it is the most well-accepted stimulus pattern for osteocytes in literature [43,44]. The external magnetic flux density (*B*) results in a torque on the magnetic beam since the direction of magnetization (*m*) is placed perpendicular to the field direction. The magnetization of the beam is 48 kA/m [45]. The resulting magnetic torque (*T*) is described by Eq 1.

$$T = m \times B = mBsin\theta \qquad (1)$$

## c) Finite element simulations and analysis

To determine the shear stress that is applied to the cells and to understand how the results can be made reproducible, a set of fluid-structure interaction simulations were performed in ANSYS Workbench 17.1. These were repeated for different deflections with a gap height of 50 μm. The results of these simulations can be seen in Figs 4 and 5. A 3D geometry was constructed as shown in Fig 2E in ANSYS Workbench 17.1 with Transient Structural and Fluent components coupled together. A tip force was applied to the magnetic beam to match the average deflection seen in experiments. Note that the devices are all manually fabricated where small variations can result in large changes to the deflection profile of the beam. This simulation uses an average observed deflection, but this can be higher or lower for individual experiments resulting in higher or lower shear stress values. The magnetic physics were not

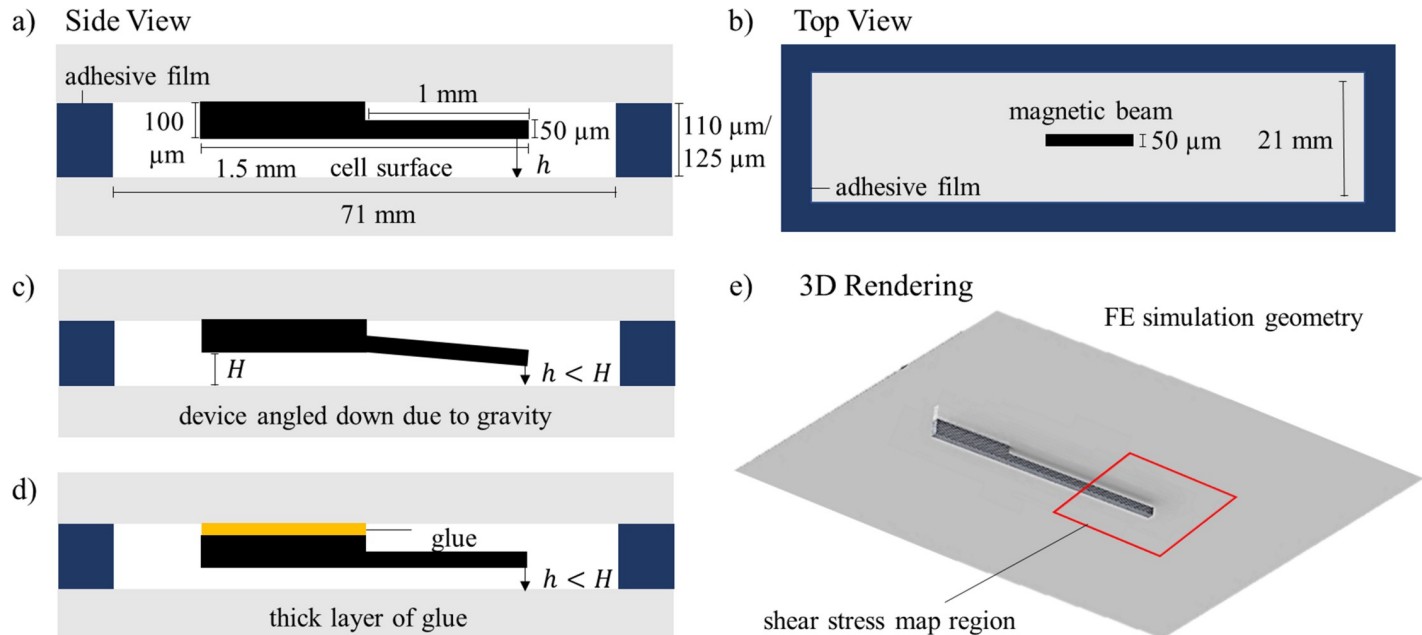

**Fig 2. Schematics showing the magnetic actuator.** a) Side view of the fully assembled device where the magnetic beam sits a distance h above the cell surface, b) top view of the device is shown with the adhesive film border on the bottom slide and the magnetic device glued to the top glass slide. c) and d) show scenarios where the gap height h can be coincidentally reduced either due to the device being angled downwards due to gravity (c) or as a result of thick spacer or a thick layer of glue pushing the whole structure downwards (d). e) A 3D rendering of the geometry is shown. This is the geometry used in the finite element analysis. The red square shows the region where shear stress calculations were performed.

modelled here since magnetic actuation is only used to deflect the beam, which is easily observed experimentally, and therefore a model is not necessary to determine beam deflection or other parameters for estimation of the shear stress. A sinusoidally oscillating force is applied to the beam which induces motion in the fluid. The resulting velocity data from the fluid domain was extracted from CFD-Post for the 1 mm square shown in red in Fig 2E. For a 2D geometry, a spatial gradient of the velocity ($u$) data can be used to generate the shear stress ($\tau$) results as shown in Eq 2 where $\mu$ is the fluid viscosity.

$$\tau(z) = \mu \frac{\partial u}{\partial z} \tag{2}$$

For our 3D geometry with all 3 components of velocity, we need to use a spatial gradient of the velocity vector, which results in a 2nd order stress tensor as seen in Eqs 3 and 4 where the diagonal elements correspond to the normal stresses and the rest are shear stress components.

$$\tau(\vec{u}) = \mu \nabla \vec{u} \tag{3}$$

$$\nabla \vec{u} = \begin{vmatrix} \dfrac{\partial u_x}{\partial x} & \dfrac{\partial u_x}{\partial y} & \dfrac{\partial u_x}{\partial z} \\[2mm] \dfrac{\partial u_y}{\partial x} & \dfrac{\partial u_y}{\partial y} & \dfrac{\partial u_y}{\partial z} \\[2mm] \dfrac{\partial u_z}{\partial x} & \dfrac{\partial u_z}{\partial y} & \dfrac{\partial u_z}{\partial z} \end{vmatrix} \tag{4}$$

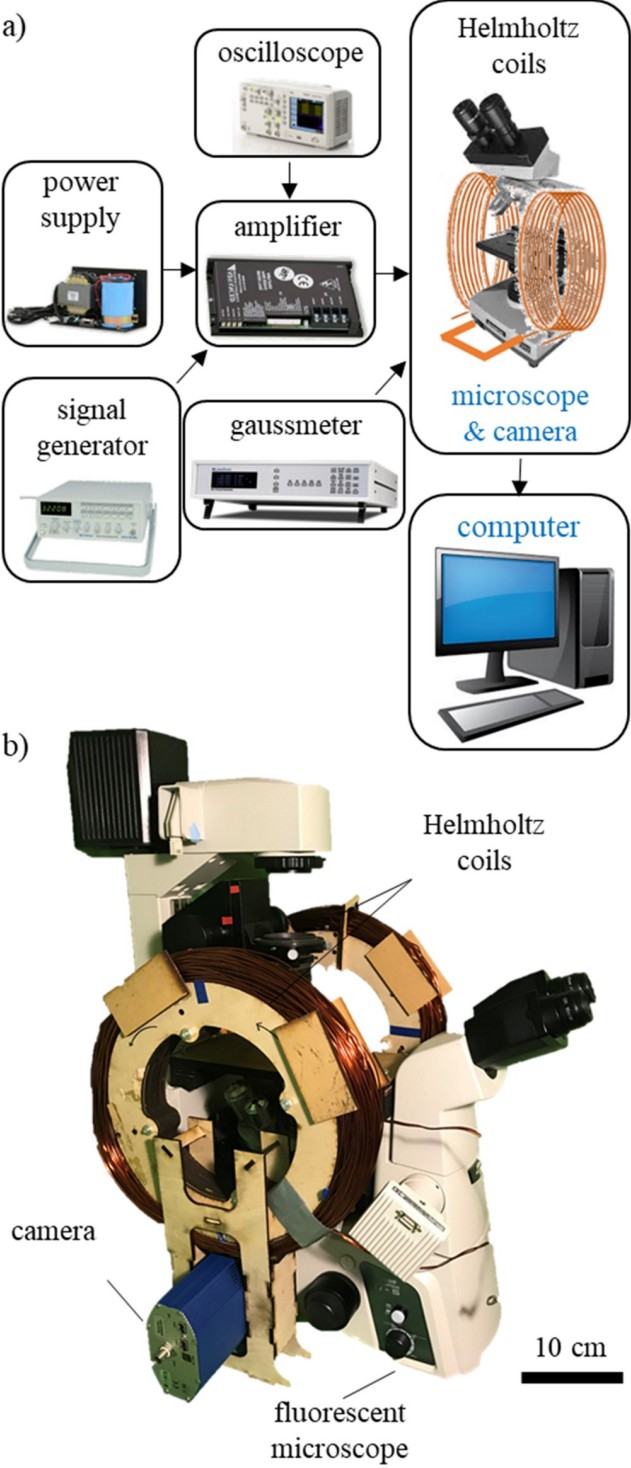

**Fig 3. Experimental setup to test magnetic actuator.** a) A schematic of the experimental setup is shown, b) the actual coils when placed around the microscope are shown. The 1D Helmholtz coils are specifically designed to fit around a fluorescent microscope. The coils are connected to an amplifier and power supply for current generation.

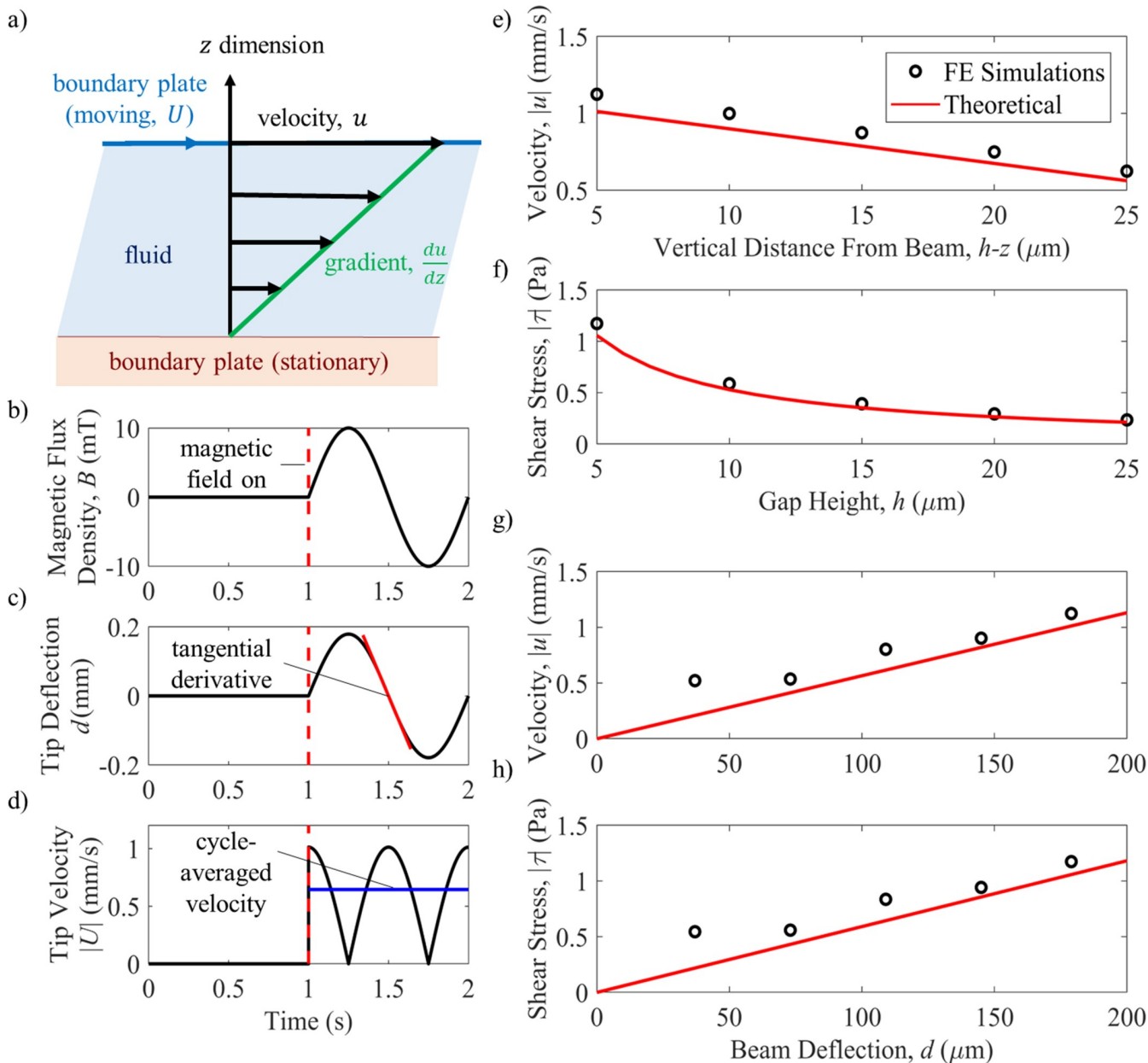

**Fig 4. Simulation results for magnetic beam flow generation.** a) 2D Couette flow principle; b) magnetic flux density generated by the coils (shown for 2 s for posterity but this takes place over several minutes); c) the resulting theoretical deflection profile of the beam if the magnetic flux density in (b) is applied. This is also the profile used in the finite element analysis and the tangential derivative is used to determine the theoretical velocities shown in parts (e) and (g). d) The theoretical beam tip velocity magnitude as well as the cycle averaged velocity is shown for the deflection profile seen in (c). e) and g) are plotted using Eq 7 with theoretical velocities determined using the tangential derivative of the deflection profile and the numerical velocities determined from the finite element simulations. f) and h) show the shear stress with varying gap heights and beam tip deflection based on Eq 6 using theoretical and numerical velocities.

When all components of the shear stress tensor were plotted, we observed that the results of the tensor are asymmetric, and this is likely a result of the vortices created by the beam oscillation and that $\tau_{xz}$ is the largest component of the shear stress. This is in line with the design of our device where the magnetic beam is placed a certain gap height above the cell surface and oscillates in the $x$ direction resulting in a large spatial gradient. Finding the principal stresses of an asymmetric tensor is computationally intensive [46]. For our purposes, it is sufficient to

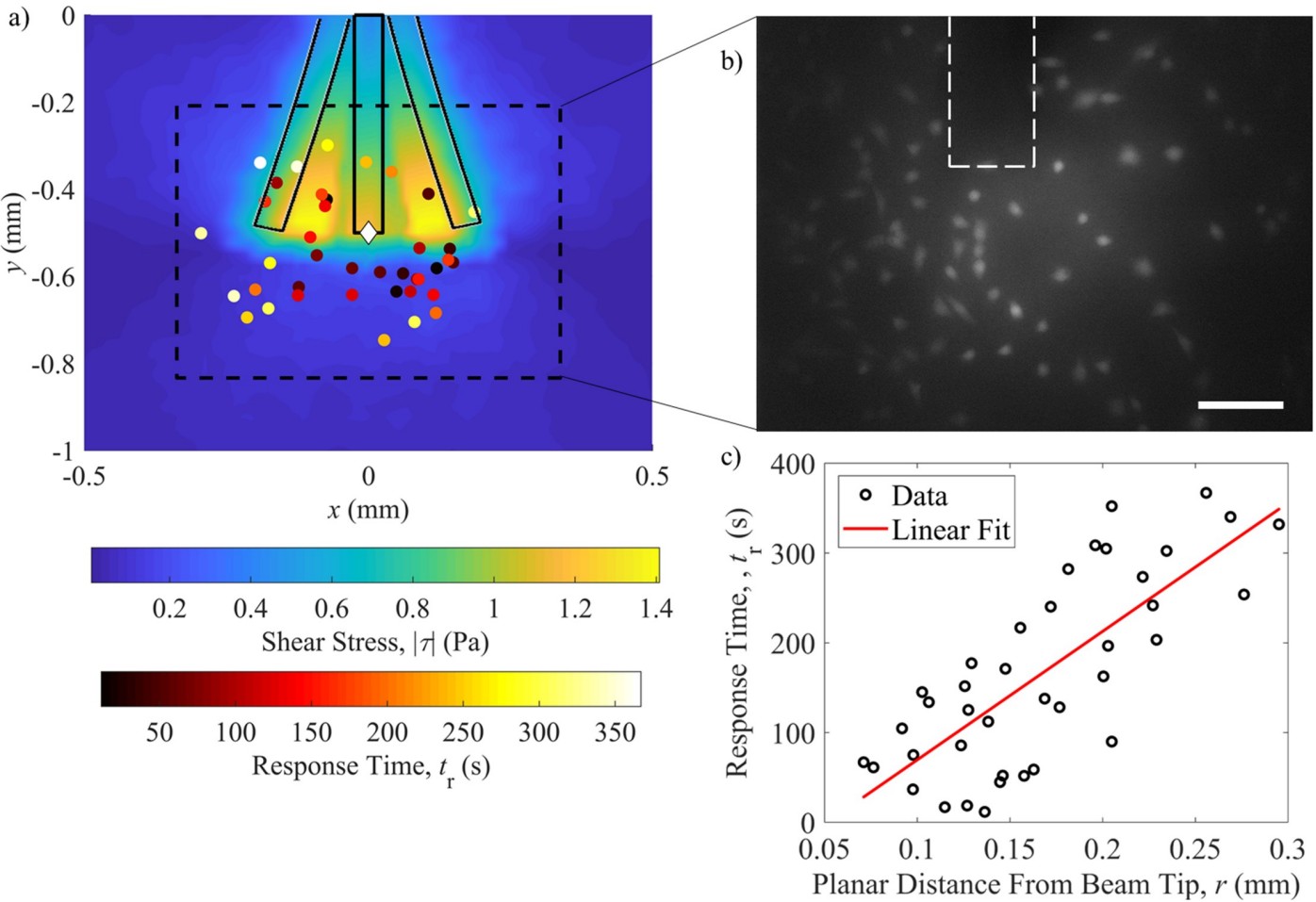

**Fig 5. Osteocyte calcium response under magnetic actuator stimulation.** a) The shear stress map obtained using Eq 6 with velocities obtained from finite element simulations is overlaid with the locations of all responding cells from multiple trials that are coded with the time it takes to respond. The dotted line shows the typical viewing window in experiments and the range of the beam oscillation is also overlaid on the map. b) A representative image obtained of the Fura-2 AM stained cells along with the beam is shown which corresponds to the dotted lines on the shear stress map. Scale bar = 50 μm. c) A plot of the cell response time vs distance from the tip of the beam with a linear regression performed is shown, showing an increase in response time for cells further away from the stimulated region. A total of 3 trials and 39 cells were recorded, with an $r^2$ value of 0.59 for the linear regression.

conclude that all other elements are negligible and concentrate on the $\tau_{xz}$ component as the primary shear stress component leading to cell stimulation.

$$\tau_{xz} = \mu \frac{\partial u_x}{\partial z} \tag{5}$$

### d) Analytical model

A Couette flow model is used as a simplified model of the system fluid dynamics. The Couette flow model is the flow of viscous fluid between two infinite plates separated by a distance $h$ with one plate moving at a velocity $U$ and the other plate held stationary as shown in Fig 4B. The shear stress for this simplified model is determined by Eq 2 (the spatial gradient of the velocity). The geometry of motion in the experimental setup of this paper differs from the Couette assumption in two main ways: 1) oscillating flow generated due to the back and forth

motions and 2) edge effects of the beam. A correction factor for the oscillatory Couette flow is found in Nalim *et al* [47]. However, since the Reynolds number for our flow is very small (with a peak of around 1.0), the oscillating flow correction is found to have negligible impact on the shear stress and does not need to be accounted for. Regarding the infinite plate assumption in the Couette flow model, our beam has defined edges and we see from the finite element model that some of the largest shear stress values occur near the edges of the beam. It has been shown in the past that edge effects increase the shear stress locally [47], and so we investigated the accuracy of the simple Couette flow model (Eq 6) for our setup to obtain the shear stress values plotted in Fig 4F and 4h. Here, $U$ is the plate velocity (or the tip velocity of the beam in our specific case), $h$ is the gap height and $z$ is the vertical distance from the stationary wall. The velocities used in the calculation of the shear stress using this model are determined using 1) finite element simulations and 2) theoretical velocities determined based on the tangential derivative of a sinusoidal wave of the beams' deflection profile as shown in Fig 4C. The theoretical velocity profile of the beam is also illustrated in Fig 4D. The fluid velocity at different vertical distances from the beam (as shown by Eq 7) is plotted in Fig 4E to show that the analytical Couette flow model and numerical simulations are in agreement and thus accurately captures the dominant fluid effects. The in-plane component of the fluid velocity is also plotted versus varying beam tip deflections in Fig 4G.

$$\tau = \frac{\mu U}{h} \tag{6}$$

$$u(z) = \frac{\mu U z}{h} \tag{7}$$

### e) Cell culture

MLO-Y4 osteocytes (courtesy of Dr. Bonewald, Indiana University School of Medicine) are cultured in growth media composed of 2.5% calf serum (CS, Gibco, USA), 2.5% fetal bovine serum (FBS, Gibco, USA), 1% penicillin-streptomycin (PS, Gibco, USA), and 94% Alpha Minimum Essential Medium (MEM) (WISENT, Canada). Cells are seeded during passage 29 at $10^5$ cells per 100 mm diameter collagen-coated (0.15 mg/ml Type I collagen (Corning, USA)) culture dishes and expanded until they achieve 80% confluency. The cells are then transferred onto collagen-coated experimental slides (75x25 mm) at a density of 500k cells per slide for overnight incubation till they reach 80% confluence again before imaging. MLO-Y4 cells are passaged between P29 and P35 while maintained in an incubator at 37 ˚C and 5% $CO_2$. Cell death was quantified using Trypan Blue Stain (Sigma-Aldrich, USA) and counted under a standard light microscope.

### f) Intracellular calcium imaging

Calcium imaging protocols are based on previously existing studies in literature [48–50]. Briefly, MLO-Y4 cells are stained with Fura-2 AM intracellular calcium dye (ThermoFisher Scientific, USA) for 45 min at room temperature in darkness. After rinsing with phosphate-buffered saline (PBS, Sigma-Aldrich, USA) and resting on a heated imaging stage for 15 min, experimental slides seeded with stained cells are imaged by a Nikon Eclipse fluorescence microscope for 1–2 minutes before the magnetic field is turned on to oscillate the beam for up to 10 minutes. During experiments, cells are seeded in regular growth media supplemented with 4.6 mg/mL Dextran (500k MW) (Sigma–Aldrich) to achieve the needed shear stress value without significantly increasing the size of the beam. This results in an increase in the viscosity of the media which is directly proportional to shear stress. It has been previously shown that

the addition of Dextran to flow experiments using MLO-Y4 osteocytes does not affect their
calcium response [24]. Fluorescence signals are read, and a ratio between signals produced
from exposure to 340 nm and 380 nm wavelength light is used to generate the calcium
response curves. A calcium response is quantified as having 2-fold-change or greater com-
pared to baseline average response peak magnitudes measured in the initial 2 minutes of non-
stimulated cells.

## Results and discussion

We see from the analytical and numerical results of the shear stress values with different gap
heights that large shear stress values are only obtained for very small gap heights (below
10 μm) and large beam tip deflection (greater than 150 μm). In Fig 5A, we have plotted the
maximum shear stress map resulting from the oscillation of the beam over 10 cycles (the flow
has stabilized within 1 cycle as differences between the 1 cycle and 10 cycle simulations appear
negligible) in the 1 mm square region of interest around the beam tip, along with all respond-
ing cells recorded from multiple experimental trials. A sample microscope view of the beam
and osteocytes is shown in Fig 5B. The maximum shear stress occurs in small areas around the
beam oscillation which we refer to as the 'local stimulation region' (LSR). We see that cells in
and around the LSR are stimulated. This is confirmed to be a response due to the magnetic
actuator, as cells seeded in the magnetic field without the actuator elicited no quantifiable cal-
cium response. We also observed cells being stimulated outside of the LSR where the shear
stress magnitude was below the threshold required for cell stimulation. We have two hypothe-
ses as to why this is the case; first, this could be the result of some form of cell response due to
prolonged low magnitude shear stress, or secondly, a release of signalling factors from the
stimulated cells in the LSR cross talk with cells outside the LSR that lead to their response
through intercellular communication. Also plotted here are the results of stimulated cells col-
our coded with a time stamp. It is observed that the response time of the cells is correlated
with the distance of that cell from the tip of the beam (Figs 5C and 6A). The linear coefficient
between these two variables is calculated to be 0.59, demonstrating a trend between distance
and response time. We believe a higher coefficient can be achieved once we adopt the experi-
mental setup in an enclosed microfluidic device, where we will have more control over the gap
height between the osteocytes and the magnetic beam, as well as the reproducibility of device
fabrication and thus the setup. This corroborates our earlier prediction that gravity is pulling
down the tip of the beam resulting in higher shear stress levels near the tip and lower shear
stress levels in the remaining LSR. The effect of gravity on the beam was also visually observed
on multiple devices. The detailed calcium response of three cells is shown in Fig 6B.

    As established in the literature, signals propagate within an osteocyte network through key
molecules such as ATP and calcium. It is interesting to note that despite the relatively far dis-
tance between the LSR and responding cells further away (100–250 μm) from the magnetic
actuator, previous work has reported that extracellular vesicles could play a key role in deliver-
ing signals at a distance [18,48]. Since *in vitro* studies using cell indentation tools have shown
it is difficult for calcium signals alone to propagate intracellular signalling beyond its neigh-
bouring cells [17], we speculate that exosomes could be a potential explanation for responding
cells outside of LSR. Furthermore, first peak response time of up to 300 s has been observed
from distant cells. As this time is much longer than standard calcium fluctuation response
time due to mechanical stimulation [51,52], it can be implied that cellular response seen at this
time scale is due to signal propagation from previously stimulated cells. However, future
experiments involving fluorescent tracing of signal molecules will be required to confirm this
hypothesis. Interestingly, except the time of initial response, there was no distinct difference in

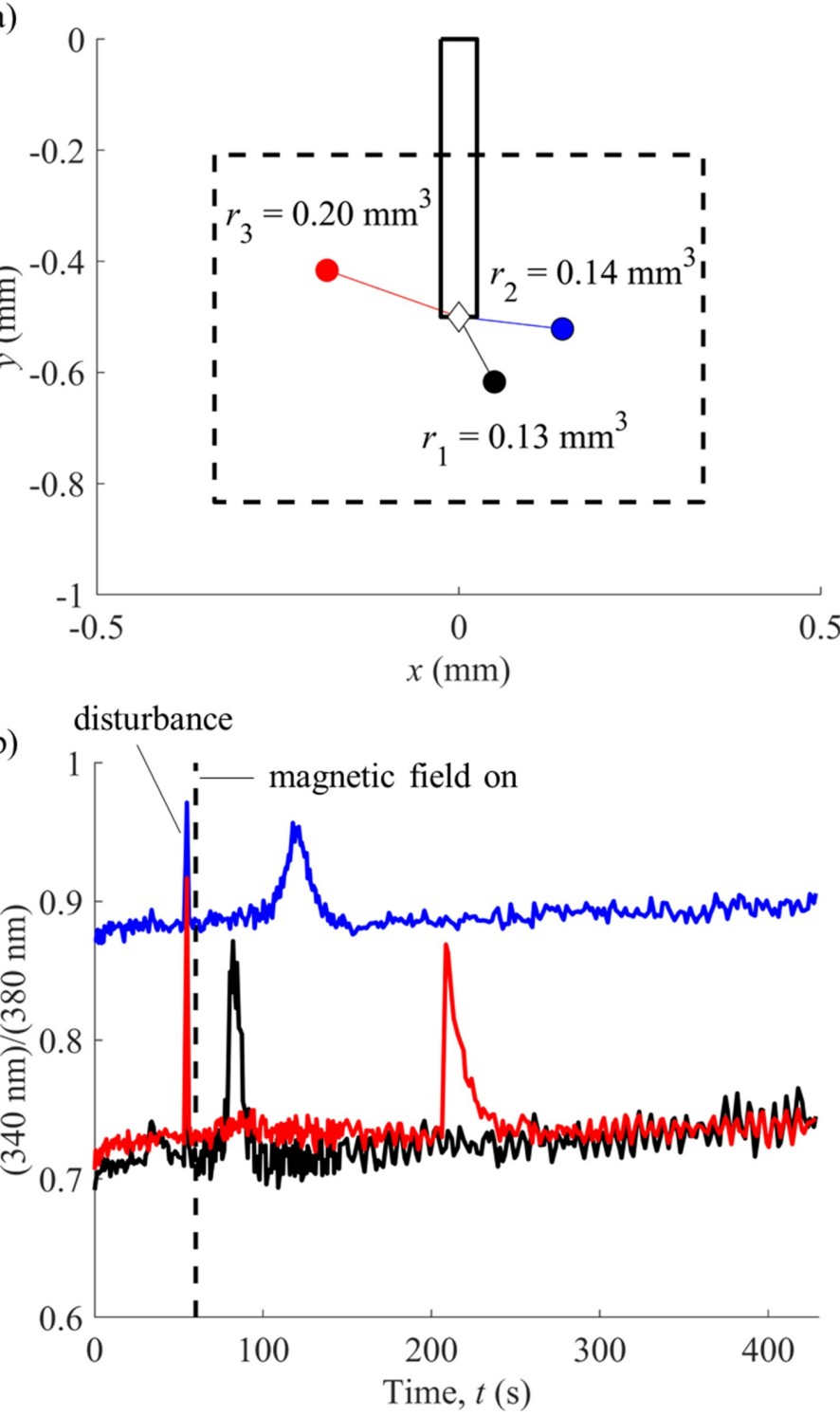

**Fig 6. Sample calcium response curves resulting from magnetic actuation.** a) Three cells are plotted with respect to their positions from the tip of the beam, positioned in regions with different shear stress values; b) the calcium response of the corresponding cells from part (a) are shown.

other response characteristics such as peak response magnitude and frequency of multi-peak response between mechanically stimulated osteocytes and osteocytes outside of the LSR with a registered calcium response. However, there was a slight, statistically non-significant trend towards higher response rate closer to the magnetic actuator (as can be seen by density of dots in Fig 5C). The above similarity in response magnitude is different from previous studies using cell membrane indentation technique, where a decrease in response magnitude was observed between stimulated and neighbouring non-stimulated cells [17]. With a prolonged stimulation time, it is possible that the concentration of signalling molecules increased to a threshold level capable of generating a comparable cellular response as fluid shear stress [53]. As there is an inherent difference between types of forces applied to the cell during fluid shear stress vs. cell membrane indentation, it is difficult to draw appropriate conclusions.

## Conclusions

A device design is proposed and fabricated in order to locally stimulate cells. The device was employed experimentally, and cells under direct beam oscillation induced shear stress were found to respond with an intracellular calcium concentration increase. A set of finite element simulations were performed in order to obtain a shear stress map and a small LSR was found at a gap height of 5 μm. Over time, cells outside the LSR also respond. We postulate that this could be the result of communication between cells from the LSR or due to prolonged application of low magnitude shear stress. Future experiments can be made more reproducible by controlling the gap height more precisely in device fabrication. Another easy way of increasing the shear stress is to further increase the viscosity of the fluid which has been shown to increase the shear stress on the cells [54]. This is a straightforward path to achieve higher levels of stimulatory shear stress observed in bone tissue during loading [55]. Future studies will aim to place the beam inside microfluidic channels in order to do more in depth molecular analysis. A microfluidic device will allow for future studies of osteocyte network signalling with physiologically accurate localized shear stress gradient.

## Supporting information

**S1 Data. Calcium response data.** Complied data showing calcium response from the three experimental trials.
(XLSX)

**S1 Video. Local stimulation of osteocytes using a magnetically actuated oscillating beam.** Visualizing the results presented in the paper.
(MP4)

## Acknowledgments

The authors would like to thank Vishal Gupta for help with construction of the coil system used during experiments.

## Author Contributions

**Conceptualization:** Onaizah Onaizah, Liangcheng Xu, Kevin Middleton, Lidan You, Eric Diller.

**Data curation:** Onaizah Onaizah, Liangcheng Xu.

**Investigation:** Onaizah Onaizah, Liangcheng Xu.

**Methodology:** Onaizah Onaizah, Liangcheng Xu, Kevin Middleton.

**Project administration:** Lidan You, Eric Diller.

**Supervision:** Lidan You, Eric Diller.

**Visualization:** Onaizah Onaizah, Liangcheng Xu.

**Writing – original draft:** Onaizah Onaizah, Liangcheng Xu.

**Writing – review & editing:** Onaizah Onaizah, Liangcheng Xu, Kevin Middleton, Lidan You, Eric Diller.

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
