## [Decision Letter · Decision Letter 0]

30 Mar 2020

PONE-D-20-06886

Local Stimulation of Osteocytes Using a Magnetically Actuated Oscillating Beam

PLOS ONE

Dear Dr. You,

Thank you for submitting your manuscript to PLOS ONE. After careful consideration, we feel that it has merit but does not fully meet PLOS ONE’s publication criteria as it currently stands. Therefore, we invite you to submit a revised version of the manuscript that addresses the points raised during the review process.

We would appreciate receiving your revised manuscript by May 14 2020 11:59PM. To enhance the reproducibility of your results, we recommend that if applicable you deposit your laboratory protocols in protocols.io, where a protocol can be assigned its own identifier (DOI) such that it can be cited independently in the future. For instructions see: http://journals.plos.org/plosone/s/submission-guidelines#loc-laboratory-protocols

We look forward to receiving your revised manuscript.

Kind regards,

Jose Manuel Garcia Aznar

Academic Editor

PLOS ONE

Journal Requirements:

Reviewers' comments:

Reviewer's Responses to Questions

**Comments to the Author**

1. Is the manuscript technically sound, and do the data support the conclusions?

Reviewer #1: Partly

Reviewer #2: Partly

Reviewer #3: Partly

2. Has the statistical analysis been performed appropriately and rigorously? 

Reviewer #1: I Don't Know

Reviewer #2: No

Reviewer #3: N/A

3. Have the authors made all data underlying the findings in their manuscript fully available?

Reviewer #1: Yes

Reviewer #2: Yes

Reviewer #3: Yes

4. Is the manuscript presented in an intelligible fashion and written in standard English?

Reviewer #1: Yes

Reviewer #2: Yes

Reviewer #3: Yes

5. Review Comments to the Author

Reviewer #1: This paper describes the fabrication of a magnetically actuated oscillating beam device for the local cellular stimulation. They validate their device by measuring Ca production from osteocytes upon stimulation. The paper is interesting, and they have performed a thorough characterization of their device; however, there are too many issues that need to be addressed before the manuscript is ready for publication. My recommendation is that the manuscript deserves a major revision in its current form, and it should be re-written and re-submitted.

If the authors are prepared to undertake the work required, I would be pleased to reconsider my decision, although this does not imply that the paper will be accepted with certainty in the event that the authors decide to revise and resubmit the work.

1. This manuscript needs a revision for grammar and clarity: e.g., fluorescent microscope (fluorescence microscope). The magnetic physics was (The magnetic physics were) The Couette model consists of flow….., The fluid velocity at different vertical distances from the beam (as shown by Eq. 7) are plotted (is),…. etc.

2. Different formatting in the references is used. Please use a uniform format. See (4,5) and (9),(10) for instance.

3. In the introduction section a brief revision about optical, electrical and magnetic non-contact cell manipulation should be included to give the readers a better understanding of the field.

4. The introduction is too short to give a good feedback about the current state-of-the art. The talk about AFM but, in addition, a description about the use of magnetic and optical tweezers in order to exert controlled mechanical stimulation on cells should be included.

5. Figure 1f. How many replicas were performed?

6. There is an error in Figure 2 caption. There is not Figure 2f. They are repeating the same Figure 1f caption.

7. The authors say: “However, the magnitude of the shear stress must be greater than 0.5Pa to result in cell stimulation(6)”. But in ref 6 at 1Hz, differences in cell stimulation compared untreated controls takes place after 2h of stimulation. So timing is important, and they authors of this current manuscript should include an explanation about the exposition times used in their work (10 min for the intracellular Ca imaging) and the rapid cellular response observed compared to other works. Why?

8. Why the fluid viscosity used in equation 2 is constant? Can it vary depending on the Ca production upon stimulation? The authors say: “a release of nutrients and chemicals from the stimulated cells”. So, does the medium change in composition depending on the stimulation? Have they calculated the viscosity of the media containing 4.6 mg/mL Dextran (500k MW) in their calculations??? An explanation is needed.

9. The authors say: “a release of nutrients and chemicals from the stimulated cells” Why a cell is going to release nutrients to the neighboring cells? Do neighboring non-stimulated cells scavenge for other nutrients to support their metabolism? An explanation is needed.

10. The authors say: Despite the relatively far distance between the LSR and responding cells further away (100-250 μm) from the magnetic actuator, extracellular vesicles could play a key role in delivering signals at distances beyond the immediate surroundings of the LSR(18,29). They could collect exosomes from non-stimulated and stimulated cells and running a proteomic analysis to validate their hypothesis. Also, to know by a simple western blotting if the number of exosomes is higher upon stimulation. If not, I do not understand why that sentence in this manuscript.

11. In Figure 5b scattered cells are shown. Do you need have confluent cells as in a physiological media to draw conclusions? Does the cell number play a role in Ca response? Experiments with different cell numbers are needed.

Reviewer #2: The authors have engineered a microfluidic device where mechanical stimulations can be applied on cells seeded in a monolayer on the bottom of the devices. For this a magnetically actuated oscillating beam was developed to generate shear stresses on the cells thanks to the oscillatory movements of the beam. Microfluidic devices are complex engineered system that can be compromised at any step of their fabrication which make this work notable. However some concerns regarding the validation can be raised.

1- One major comment on this study, is the few statistical analysis provided regarding the capacity of the devices to stimulate osteocyte-like cells (MLO-Y4) and to induce intracellular calcium signalling. Indeed, many details are lacking specifically in the materials and methods section, ie the number of independent experiments that have been performed and that worked, and if any statistical tests were performed to compare the results of calcium signalling. In the current state of the article, the results are presenting data from only 1 trial since 39 cells were recorded in a video from a single run and are shown in the figure 5 which does not represent the 3 trials that are informed in the caption. In addition for the figure 5c the coefficient of regression is only available in the caption and is not discussed.

2- Osteocytes are the orchestrator of bone homeostasis and are mechanosensitive cells. What is the minimum of flow subjected on the osteocytes that can induce a response? What is the range of shear stress in the lacunocanalicular network in healthy condition?

3- In the FEA, there is no indication of any turbulence induced in the fluid outside of the LSR due to the oscillatory movement of the beam. What str the authors thought about the flow induced outside this area? Could this induce the oscillation of calcium observed in cells far from the beam?

4- It is not clear what is the timescale to have a stable flow. And how this can be compared to other devices inducing fluid flow to study calcium signalling in vitro?

5- Additionnal introduction on microfluidics and the fluid flow induced would allow to understand what are the advantages of this device compared to others.

Minor comments

1- In the Introduction, there is a confusion in citing McGarrigle et al 2016. In this article, the authors did not quantify intracellular calcium signalling and no fluid flow systems were used. Is there a confusion with the article from Deepak et al 2017? Please make sure all the other references are cited appropriately.

2- Some details are lacking in the materials and methods ie the passage of the cells when seeded on the devices.

3- What is the maximum duration that the system was ran to induce an oscillatory shear stress?

4- It would improve the manuscript to add in the discussion a sentence about the absence of control conditions where the cells are subjected to the magnetic field and not to shear stress.

Reviewer #3: Figures are schematic, illustrative and well captioned. However, they are quite pixelated (especially figures 1, 2 and 3). Graphics could be acceptable in smaller size, but it would be better to see them in higher quality.

Line 112: “shear stress must be greater than 0.5 Pa to result in cell stimulation”. This is good data to set the experiment but a reference is missed regarding physiological value (or range of values) of shear stress that osteocytes may be exposed to in lacuno-canicular system. Indeed, in lines 19-21 (abstract) you claim that the beam generates the fluid shear stress encountered in vivo but you did not indicate that value. Related to the former comment, in line 61: you set a frequency of 1Hz in order to apply fluid shear stress to the cells. Have you tested different frequencies to get a wider scope of shear stress?

Line 174: Cell density is reported to be a key parameter in 'in vitro' models. The density seeded over the experimental slides is (500k cells / (7.5cm*2.5cm); (line179)) ≈ 27.000 cells/cm2. It is a value that is not compared with physiological osteocyte density in murine lacuno-canicular system. Did you check results of calcium responses with different cell densities?

Line 186: you have stimulated the culture for 10 min. Afterwards you indicate that “With a prolonged stimulation time, it is possible that the concentration of signaling molecules increased to a threshold level capable of generating a comparable cellular response as fluid shear stress”. Have you tested your experiment for longer times to check this possibility?

Line 187: “growth media supplemented with Dextran to achieve the needed shear stress without significantly increasing the size of the beam”. It is not completely clear the reason why Dextran is needed to achieve that purpose. It may be also helpful to show a fluorescent picture of the culture.

You define LSR (local stimulation region) as areas in and around the beam oscillation (line 199). Figure 5, both images “a” and “b”, are useful pictures that ease the understanding of the results you provide. It would be helpful, though, to indicate on both figures what you call LSR if it helps to distinguish between cells inside and outside LSR (lines 199-206).

6. PLOS authors have the option to publish the peer review history of their article (what does this mean?). If published, this will include your full peer review and any attached files.

Reviewer #1: No

Reviewer #2: No

Reviewer #3: No

---

## [Author Response · Author response to Decision Letter 0]

14 May 2020

Dear Reviewers,

Based on the comments provided by the reviewers, we have modified our manuscript to provide clarification on the issues raised. These changes focused primarily on justifying the shear stress levels that were generated and how we quantified these values. We also provided an explanation regarding the duration of the experiment, as well as the confluence level used during these experiments. We have also provided clarification for the number of experimental trials that we used to collect our data. We also added a paragraph to the introduction in order to highlight other techniques used for cell manipulation and their advantages and limitations. You can find our detailed response in the attached Response to Reviewer document.

Sincerely,

Authors

---

## [Decision Letter · Decision Letter 1]

15 Jun 2020

Local Stimulation of Osteocytes Using a Magnetically Actuated Oscillating Beam

PONE-D-20-06886R1

Dear Dr. You,

We’re pleased to inform you that your manuscript has been judged scientifically suitable for publication and will be formally accepted for publication once it meets all outstanding technical requirements.

Kind regards,

Jose Manuel Garcia Aznar

Academic Editor

PLOS ONE

Additional Editor Comments (optional):

Reviewers' comments:

Reviewer's Responses to Questions

**Comments to the Author**

1. If the authors have adequately addressed your comments raised in a previous round of review and you feel that this manuscript is now acceptable for publication, you may indicate that here to bypass the “Comments to the Author” section, enter your conflict of interest statement in the “Confidential to Editor” section, and submit your "Accept" recommendation.

Reviewer #1: All comments have been addressed

Reviewer #3: All comments have been addressed

2. Is the manuscript technically sound, and do the data support the conclusions?

Reviewer #1: Yes

Reviewer #3: Yes

3. Has the statistical analysis been performed appropriately and rigorously? 

Reviewer #1: Yes

Reviewer #3: N/A

4. Have the authors made all data underlying the findings in their manuscript fully available?

Reviewer #1: Yes

Reviewer #3: Yes

5. Is the manuscript presented in an intelligible fashion and written in standard English?

Reviewer #1: Yes

Reviewer #3: Yes

6. Review Comments to the Author

Reviewer #1: The authors have satisfactorily addressed all my concerns. I think that the manuscript is ready for publication. There are still a three issues with their model which deserve more investigation as I proposed but as the authors say they will include those in the next generation of their new encased microfluidic devices.

I think that as it is the manuscript contains enough new results which are interesting for the scientific community working on the field.

Reviewer #3: I consider the authors have addressed the comments raised in this round of review. The introduction frames deeper the current state of the art. Thus, the obtained results are now clearly understood and outlined.

It is also noteworthy the improvement in written English quality. However, from my point of view there are some redundant sentences and the choice of words has room for improvement. In my opinion, a minor revision is needed. Here, I would like to present some optional recommendations that might improve the writing style:

- L 62 "While local stimulation of cells has been attempted in the past [19], no study has attempted to quantify". You wrote twice "attempted", i may use another word as "no study has succeeded in.."

- L 68: "allow for cutting, injecting and stimulation of individual cells". It might be grammatically more accurate "allow to cut, inject and stimulate individual cells".

- L 70: "They are, however, less likely to damage cells". I consider that it sounds informal and something that is not sure, but non-contact methods are indeed less harmful to cells. Thus, I would write that in a different but similar way "However, -they/those methods- are less harmful to cells".

-L 210: "while maintained in an incubator at 37 ˚C". They are indeed "incubated at" this T and CO2%.

-L 213: "Calcium imaging protocols are based on". As far as I understood, you are relating a single calcium protocol in that paragraph. Thus, it would be better to write that in singular "Calcium imaging protocol is base on.."

-L 228-230. You are splitting the provided information by adding more info in brackets. It may be clearer to reorder "...from the oscillation of the beam over 10 cycles in the 1 mm 230 square region of interest around the beam tip, along with all responding cells recorded from multiple experimental trials. The flow has stabilized within 1 cycle as differences between the 1 cycle and 10 cycle simulations appear negligible. A...".

-L 271: "(as can be seen by density of dots in Fig 5c)." better "see dots density in Fig 5c".

I would like to congratulate you on the effort you make in this round of revision.

7. PLOS authors have the option to publish the peer review history of their article (what does this mean?). If published, this will include your full peer review and any attached files.

Reviewer #1: No

Reviewer #3: No

---

## [Editor Report · Acceptance letter]

18 Jun 2020

PONE-D-20-06886R1 

Local Stimulation of Osteocytes Using a Magnetically Actuated Oscillating Beam 

Dear Dr. You:

I'm pleased to inform you that your manuscript has been deemed suitable for publication in PLOS ONE. Congratulations! Your manuscript is now with our production department. 

Kind regards, 

on behalf of

Dr. Jose Manuel Garcia Aznar 

Academic Editor

PLOS ONE